# Anti-Inflammatory and Antioxidant Properties of Tart Cherry Consumption in the Heart of Obese Rats

**DOI:** 10.3390/biology11050646

**Published:** 2022-04-23

**Authors:** Ilenia Martinelli, Daniele Tomassoni, Vincenzo Bellitto, Proshanta Roy, Maria Vittoria Micioni Di Bonaventura, Francesco Amenta, Consuelo Amantini, Carlo Cifani, Seyed Khosrow Tayebati

**Affiliations:** 1School of Pharmacy, University of Camerino, 62032 Camerino, Italy; vincenzo.bellitto@unicam.it (V.B.); mariavittoria.micioni@unicam.it (M.V.M.D.B.); francesco.amenta@unicam.it (F.A.); carlo.cifani@unicam.it (C.C.); khosrow.tayebati@unicam.it (S.K.T.); 2School of Biosciences and Veterinary Medicine, University of Camerino, 62032 Camerino, Italy; daniele.tomassoni@unicam.it (D.T.); proshanta.roy@unicam.it (P.R.); consuelo.amantini@unicam.it (C.A.)

**Keywords:** obesity, heart, tart cherries, inflammation, oxidative stress, cardiovascular diseases

## Abstract

**Simple Summary:**

Obesity is a well-known condition responsible for being a risk factor for cardiovascular disease progression. The intake of bioactive phytochemicals, contained in red fruits, is attracting great attention since their benefits have been attributed mostly to their possible antioxidant properties. We aimed to assess the potential effects from the daily supplementation of tart cherries, both seeds and juice, in obese animals. Our results showed that tart cherries reduced oxidative stress and mitigated the inflammation in the hearts of obese rats. Indeed, we propose this fruit in the prevention of cardiovascular diseases related to obesity.

**Abstract:**

Obesity is a risk factor for cardiovascular diseases, frequently related to oxidative stress and inflammation. Dietary antioxidant compounds improve heart health. Here, we estimate the oxidative grade and inflammation in the heart of dietary-induced obese (DIO) rats after exposure to a high-fat diet compared to a standard diet. The effects of tart cherry seed powder and seed powder plus tart cherries juice were explored. Morphological analysis and protein expressions were performed in the heart. The oxidative status was assessed by the measurement of protein oxidation and 4-hydroxynonenal in samples. Immunochemical and Western blot assays were performed to elucidate the involved inflammatory markers as proinflammatory cytokines and cellular adhesion molecules. In the obese rats, cardiomyocyte hypertrophy was accompanied by an increase in oxidative state proteins and lipid peroxidation. However, the intake of tart cherries significantly changed these parameters. An anti-inflammatory effect was raised from tart cherry consumption, as shown by the downregulation of analyzed endothelial cell adhesion molecules and cytokines compared to controls. Tart cherry intake should be recommended as a dietary supplement to prevent or counteract heart injury in obese conditions.

## 1. Introduction

Most evidence supports a connection between obesity and cardiovascular diseases (CVD), including coronary heart disease, heart failure, hypertension, stroke, atrial fibrillation, and sudden cardiac death [1,2,3]. The cardiovascular system is structurally and functionally modified to accommodate excess body weight. Consequently, the increase in dysregulated adipokine secretion enhances inflammation and perturbates vascular homeostasis. Insulin resistance, hyperglycemia, hypertension, and dyslipidemia are recognized as concomitant risk factors in the association between obesity and CVD, and the consequences are often attributed to pro-inflammatory and pro-thrombotic conditions as well as endothelial dysfunction and platelet activation [2,3,4,5].

It has been found that feeding a diet rich in fat and carbohydrates leads to significant oxidative stress and inflammation in obese subjects [6]. Concerning the mechanisms in obese persons, the adipose tissue secretion of adipokines and cytokines or chemokines is dysregulated. These bioactive molecules participate in the regulation of appetite and energy homeostasis, lipid metabolism (tumor necrosis factor-alpha, TNF-α), insulin sensitivity (TNF-α, adiponectin, resistin, visfatin), immunity (TNF-α, monocyte chemoattractant protein-1, MCP-1 and interleukin-6, IL-6), angiogenesis, blood pressure, and hemostasis (plasminogen activator inhibitor, PAI-1) [4,7,8]. Additionally, in obesity, chronic low-grade of inflammation is a central source of oxidative stress. Mitochondrial dysfunction leads to the alteration of free radical production and fatty acid oxidation; both have been implicated in the pathogenesis of obesity and its associated risk factors [9,10]. Therefore, a diet rich in antioxidants protects the cell from free radical injury by counteracting and scavenging them [11]. For instance, cherries contain different polyphenolic compounds that have a beneficial impact on human health [12].

Considerable interest has been shown in diets enriched with natural bioactive substances and their capacity for preserving or improving cardiovascular health [13,14]. High consumption of vegetables and fruits has been directly connected with a reduced incidence of CVD [15], mostly due to the abundance and variability of bioactive composites within. Among them, anthocyanins (members of the flavonoid group) have emerged as beneficial in animal and human studies [16,17]. As reviewed by Mazza [18], many studies have revealed that anthocyanins show an extensive variety of biological actions, such as antioxidant [19,20], anti-inflammatory [21,22], and anti-carcinogenic activities [23]; induction of apoptosis [24]; and neuroprotective effects [25,26]. Moreover, anthocyanins show a diversity of properties on blood vessels [27,28] and platelets [29] that may diminish the incidence of coronary heart disease [30]. Interestingly, anthocyanins may decrease the cardiovascular risk associated with endothelial dysfunction and inflammatory responses to a typical high-fat “Western” meal [31].

Because obesity is characterized by both a chronic state of oxidative stress and low-grade inflammation, here we explored the cardiac potential alterations in a diet-induced obesity (DIO) animal model, in which rats were fed a high-fat diet (HFD) and then the influence of tart cherry seed and juice intake were detected, assessing oxidative stress and inflammatory markers.

## 2. Materials and Methods

### 2.1. Animal and Blood Parameters

The cardiac samples were collected from the same male Wistar rats (*n* = 44; 225–250 g) described in the paper by Micioni Di Bonaventura et al. [26]. Institutional Guidelines, conformed with the Italian Ministry of Health (protocol number 1610/2013) and associated guidelines from the European Communities Council Directive were followed. Animals were divided into: CHOW rats (*n* = 8, standard diet, 7% fat) and DIO rats (*n* = 36, HFD, 45% fat) [26]. In the DIO group, 6 rats were excluded because they were resistant [26]. The effects of *Prunus cerasus* L. supplementation were assessed in DIO animals, and the concentration of anthocyanins tested, as well as the preparation of seed powder and juice from tart cherries, were previously detailed [26,32]. The composition of both juice and seeds has been already cited elsewhere [32]. In the paper by Cocci et al. [32], the fatty acid composition of seeds was assessed. After 17 weeks of HFD, animals were sacrificed, and heart weights were recorded. Blood parameters were reported previously as well as systolic blood pressure [26,32,33,34,35,36]. DS and DJS groups presented a decrease in systolic blood pressure in comparison with DIO rats. The consumption of tart cherry counteracted only the hyperglycemia but not the hyperinsulinemia. Moreover, the *Prunus cerasus* L. diminished the triglyceride levels compared to the DIO control rats [26,32,33,34].

### 2.2. Morphological Aspects

After tissues were excised, hearts were fixed in 4% paraformaldehyde; after they were dehydrated by graded alcohols and embedded in paraffin. These samples were cut using the microtome to prepare longitudinal tissue sections 8 µm-thick. Sections were deparaffinized immersing in xylene, and rehydrated through graded alcohols, followed by staining of hematoxylin and eosin (Diapath S.p.A., Martinengo, BG, Italy, Ref. 010263), silver impregnation (Diapath S.p.A., Martinengo, BG, Italy, Ref. 010211), and Masson’s trichrome (Diapath S.p.A., Martinengo, BG, Italy, Ref. 010210). Cardiomyocyte cross-sectional area and fibrosis were measured, as described previously [37,38].

### 2.3. Western Blot and Quantification

For Western blots (WB), cardiac samples were lysed in lysis buffer, whose composition has been already detailed in [38]. After centrifugation, the supernatants were collected. Proteins were measured, separated by 8–12% SDS-PAGE, and transferred to a nitrocellulose membrane. Membranes were probed with the indicated antibodies, including anti-intracellular adhesion molecule-1 (ICAM-1), anti-vascular cell adhesion molecule-1 (VCAM-1), anti-platelet endothelial cell adhesion molecule-1 (PECAM-1), anti-endothelial-leukocyte adhesion molecule-1 (E-selectin), anti-nuclear factor kappa-light-chain-enhancer of activated B cells subunit p50 (NF-κB p50), anti-TNF-α, anti-interleukin-1β (IL-1β), anti-IL-6 (Santa Cruz Biotechnology, Inc., Dallas, TX, USA), and anti-caspase-3 (9662, Cell Signaling Technology, Danvers, MA, USA) at 4 °C overnight. β-actin (A2228, Sigma-Aldrich Co., St. Louis, MO, USA) was used as loading control. Optimal antibodies concentration was previously established [26,38]. After incubation with horseradish-peroxidase (HRP)-conjugated secondary antibodies (Bethyl Laboratories, Inc., Montgomery, TX, USA), followed by enhanced chemiluminescence (ECL) method, protein signals were measured. The densitometric analysis of bands were performed [38]. Finally, anti 4-Hydroxynonenal antibody (4-HNE) (sc-130083, Santa Cruz Biotechnology, Inc., Dallas, TX, USA) was used to assess the lipid peroxidation in cardiac homogenates. Moreover, the protein carbonyl levels were analyzed using Oxyblot kit, as detailed [38]. For 4-HNE and oxyblot, the images were quantified by measuring the intensity of the whole protein lane.

### 2.4. Immunohistochemistry and Image Analysis

Cardiac sections were processed for immunohistochemistry (IHC) analysis, as previously described [38]. The primary antibodies used for WB, were also incubated for IHC in tissues sections overnight at 4 °C: anti-ICAM-1 (sc-8439), anti-VCAM-1 (sc-8304), anti-PECAM-1 (sc-1506), anti-E-selectin (sc-14011), anti-TNF-α (sc-52746), anti-IL-1β (sc-7884), and anti-IL-6 (sc-1265) (Santa Cruz Biotechnology, Inc., Dallas, TX, USA). Optimal antibodies concentration has already been established [38]. The sections were incubated in biotinylated secondary antibody (Bethyl Laboratories, Inc., Montgomery, TX, USA) and then with VECTASTAIN ABC HRP kit, according to the manufacturer’s protocol. 3,3′-diaminobenzidine tetrahydrochloride (DAB) substrate was then applied on the sections. Both these kits were purchased by Vector Laboratories (Burlingame, CA, USA). Finally, slides were counterstained with hematoxylin, and observed under a light microscope. Cardiac pictures were captured at 40×. The mean intensity of immunostaining was recorded as previously described [26].

### 2.5. Immunofluorescence and Quantification

For confocal microscopy, the sections were incubated with antibody specific for NF-κB (p50) (sc-114, Santa Cruz Biotechnology, Inc., Dallas, TX, USA). The secondary antibody was conjugated with Alexa Fluor 594 (red) and sections were counterstained with DAPI. Slides were observed with confocal microscope (Nikon, Corporation, Tokyo, Japan). Pictures were captured and the percentage of NF-κB (p50) positive area was measured with Nikon NIS Element software.

### 2.6. Data Analysis

The statistical significance of the differences was performed with GraphPad Prism program (version 8.0) by analysis of variance (ANOVA) followed by Tukey’s post hoc test. Data represented as mean ± standard error of mean (SEM). Statistical significance was set at *p* < 0.05.

## 3. Results

### 3.1. Weight and Heart Morphology

Tart cherry seeds and juice consumption did not alter body weight or feeding performance [26,33,34]. Heart weights were measured, and no remarkable differences were found among the different experimental groups (Figure 1A). Morphological measures of the myocardium performed on the sections stained with hematoxylin and eosin exhibited an increase in the ventricular cardiomyocyte area of DIO animals compared to CHOW rats (Figure 1B,C). The consumption of tart cherries restored the area of cardiomyocytes (Figure 1B,C).

In cardiac sections of DIO rats, processed with the silver impregnation, a slight increase in the reticulin fibers was evident (Figure 2A). The quantification of the silver-stained area, expressed as a percentage, demonstrated no differences among the groups (Figure 2B). Collagen and extracellular matrix were not increased at the level of subendocardial area of the obese rats as showed in Masson’s trichrome stained sections (Figure 2C). The amounts of fibrosis were low in all the animal groups without differences between the groups (Figure 2D).

### 3.2. Oxidative Stress and Apoptosis

Previously, a decrease in oxidative stress was reported with the intake of tart cherries in the serum and liver of obese rats [33]. In accordance, the quantification of oxyblot assay and the WB analysis for 4-HNE in heart homogenates demonstrated an increase in oxidation state of proteins and lipid peroxidation, respectively, in DIO samples compared to control rats (Figure 3A,B). Furthermore, the tart cherries were able to inhibit both these conditions (Figure 3A,B). In addition, we investigated the potential modulation of apoptosis induced by diet and tart cherry supplementation. The data showed no difference in the full-length caspase-3 (35 kDa) levels, without cleaved caspase-3 (17 kDa) expression in all the samples (Figure 3C). In Appendix A, we showed the full-length WB gels (Appendix A).

### 3.3. Inflammation

As the NF-κB is a transcription factor responsible for triggering the immune response, and the most prevalent activated form of NF-κB is a heterodimer, consisting of a p50 or p52, its expression was investigated. Moreover, ICAM-1, VCAM-1, PECAM-1, E-selectin, TNF-α, IL-1β, and IL-6 levels were studied since NF-κB activation increases the expression of the adhesion molecules as well as the production of pro-inflammatory cytokines [39]. As shown in WB (Figure 4A) and immunofluorescence quantification (Figure 4B), a substantial upregulation of this transcription factor was reported in DIO rats, and those levels were remarkably lowered, both in DS and in DJS rats. We showed the full-length WB of NF-κB in Appendix A.

#### 3.3.1. Adhesion Molecules

WB, performed on heart tissue lysates, showed relevant changes in ICAM-1 and PECAM-1 expression at 90 and 130 kDa, respectively (Figure 5A,C). No difference was evident for VCAM-1 and E-selectin expressions in cardiac homogenates (Figure 5B,D). In Appendix A, we showed the full-length WB gels of these adhesion molecules (Appendix A).

In the IHC, the adhesion molecules analyzed were found expressed in the blood vessels. As shown by representative pictures of immunohistochemical staining for VCAM-1 (Figure 6A) and PECAM-1 (Figure 6B), no significant difference occurred between lean and obese animals also supplemented with tart cherry seeds or juice. It was demonstrated that VCAM-1 was present also in the cardiomyocytes (Figure 6A), as reported elsewhere [38,40].

#### 3.3.2. Cytokines

TNF-α, IL-1β, and IL-6 are the most well-recognized intermediaries of the early inflammatory reply. Inflammation consisting of enhanced cytokines levels was found in the DIO in comparison with CHOW rats. In HFD fed rats, the expressions of TNF-α (26 kDa) and IL-6 (21 kDa) were downregulated both in DS and DJS groups (Figure 7A,C). Finally, IL-1β was reduced significantly by only the association of seeds and juice (Figure 7B). In Appendix A, we showed the full-length WB gels of these cytokines (Appendix A).

The protein quantification in WB was according to the immunohistochemistry measurements expressed as mean immunoreaction intensities of the mentioned cytokines (Figure 8A,C,E). In particular, the immunoreaction for TNF-α (Figure 8B), IL-1β (Figure 8D), and IL-6 (Figure 8F) was well defined in the damaged cardiomyocytes of DIO rats. Representative pictures showed a clear reduction in TNF-α and IL-6 with seeds as well as with seeds plus juice supplementation compared to the obese DIO rats (Figure 8B,F).

## 4. Discussion

Obesity-related health complications are increasing, and the consumption of HFD is considered a major cause of these complications. Clarifying mechanisms involved in obesity-related cardiac impairments requires suitable animal models. Rodents as genetic models of obesity, such as the Zucker obese rat, the ob/ob rat, and the spontaneously hypertensive obese rat, have been studied for cardiovascular research [38,41,42]. In the present study, rats with HFD developed an obese phenotype, impaired blood parameters, and pressure, accompanied by cardiovascular alterations. The cardiovascular abnormalities presented in other diet-induced obese models were hypertension, tachycardia, reduced cardiac contractility, increased end-diastolic pressure left ventricular hypertrophy, and increased collagen deposition [43]. The progress of cardiac fibrosis has been found in many obese animal models, and it is often accompanied by diastolic dysfunction. The severity of cardiac fibrosis varies, depending on the age, species, and strain of the animals, the underlying mechanism of obesity, and the presence of concomitant pathophysiological conditions (e.g., metabolic dysfunction and hypertension) [44]. Our results on histological sections of cardiac tissue showed cardiomyocyte hypertrophy but not fibrosis in the myocardium of DIO rats. Accordingly, HFD may even be less effective in the induction of cardiac fibrotic alterations [44]. In male C57/BL6J mice, Calligaris et al. reported that feeding with an HFD for 16 months was required to develop substantial cardiac fibrosis and hypertrophy [45]. In addition, 20-week-old obese Zucker diabetic fatty/spontaneously hypertensive heart failure F1 hybrid (ZSF1) male rats showed important alterations in major systemic biomarkers of cardiovascular function without histopathological modifications in the heart [46].

Oxidative stress also causes myocardial tissue damage and inflammation, contributing to heart failure progression [47]. The signaling, mediated by the activation of inflammatory markers or NF-κB and other transcription factors as central regulators of inflammation, is the key issue to understanding oxidative stress responses in obesity [9]. Previous works have suggested an enhanced NF-κB activation in obese individuals and experimental animals fed with HFD [9,48,49]. The reply of endothelial cells to NF-κB activation and inflammation consists in the induction of adhesion molecules, promoting binding and transmigration of leukocytes, while instantaneously improving their thrombogenic potential [50]. Lee et al. [51] reported that cardiac TNF-α protein level and serum IL-6 were remarkably augmented in db/db mice and were linked with endothelial dysfunction in the coronary microvasculature. In another study, coronary endothelial dysfunction resulted from elevated plasma concentration and the expression of TNF-α and its receptor (TNFR1) in coronary arterioles of db/db mice [52]. Moreover, TNF-α gene knock-out or treatment with a TNF-α neutralizing antibody improved endothelial function of coronary arterioles in diabetic mice [52]. In our study, obesity-induced damage to the heart was associated with NF-κB activation and increased in ICAM-1, PECAM-1, TNF-α, IL-1β, and IL-6 expression levels. Collectively, our current findings in the heart support the previous study in obese Zucker rats [38] and further suggest the interplay among inflammation and oxidative stress in the obese myocardium. As summarized by Adrielle Lima Vieira et al. [53], reports indicate high levels of circulating cell adhesion molecules in obesity, especially in the presence of visceral adipose tissue accumulation [54,55,56]. Even if other studies described an intensification of VCAM-1, ICAM-1, and E-selectin in obese condition [55,56,57], and specifically in the aorta of obese Zucker rats at 15 weeks of age [58], our data displayed no differences in E-selectin and VCAM-1 in DIO compared to CHOW rats. Perhaps in obesity, the modulation of the vascular adhesion proteins expression may happen in the peripheral arteries rather than the coronary arteries [38]. Studies proposed a link between the levels of these molecules and the anthropometric markers of obesity, but they are still controversial [53]. Finally, the HFD-fed rats did not show the induction of caspase-3 activation either in the liver [33] or in the heart: this indicates that the hepatocytes and cardiomyocytes alterations were not related with cell death and/or fibrosis [33,39].

Tart cherry supplementation reduced the systolic blood pressure, glycemic values, oxidative stress, and inflammation, confirming its positive effects on the risk factors related to obesity and metabolic syndrome [59,60,61,62,63]. For instance, we have already reported that in rats fed with seeds and juice, a remarkable decrease in systolic blood pressure in obese rats was found [26,33,34]. This effect has been attributed to anthocyanins, which exhibit several biological effects, including vasodilatory capacity. In fact, the nitric oxide (NO) pathway could be responsible for the relaxation response of coronary arteries to red fruit extracts [64], and anthocyanins condensed tannin-containing fractions showed more vasodilation property than other polyphenols [17]. In hypertensive rodents supplemented with blueberry-enriched anthocyanins, a reduction in blood pressure was assessed. The anthocyanins exhibited NO-dependent vasodilation via endothelium triggered by acetylcholine. Moreover, endothelium-dependent relaxation is another vasodilator outcome derived from anthocyanins [64,65]. In clinical trials, the circulating phenolic compounds of sour cherries were able to counteract the hypertension [66]. In addition, the anti-hypertensive capacity of sour cherry could be attributed to its anti-inflammatory and antioxidant capacities [67]. Interestingly, there are two key components in the tart cherry seeds: the oleic and linoleic acids, which can protect the endothelium [68]. These fatty acids could elucidate the anti-inflammatory properties also observed in the cerebral areas of DIO rats [26]. In obese and diabetic conditions, a diet rich in oleic acid can induce the positive effect reversing the negative consequences of inflammatory cytokines [69]. Oleic acid has been proposed to protect against cardiovascular insulin resistance, improving endothelial injury in response to proinflammatory activation, and reducing apoptosis in vascular smooth muscle cells. All these properties may help to maintain plaque stability and to ameliorate the atherosclerotic process [70]. Furthermore, it has been reported that the intake of sour cherry seed kernel extract improved postischemic recovery of cardiac function during reperfusion. Moreover, other potential action mechanisms of proanthocyanidin, trans-resveratrol, and flavonoid components of the extract could be responsible for the cardioprotection in ischemic-reperfused myocardium [71]. Finally, preclinical evidence showed benefits in dietary supplementation of the cited antioxidant compounds on fat accumulation [11].

Here, we demonstrated a clear reduction in protein carbonyl levels and 4-HNE in rodents HFD supplemented with tart cherries, compared to DIO rats. In particular, the upregulation in 4-HNE because of oxidative stress was detected in numerous cardiac pathologies (e.g., diabetic cardiomyopathy). 4-HNE damages the myocardium, interfering with mitochondria and making adducts [72]. Previously, we reported not only a decrease in oxidative stress both in the serum and in the liver [33], but also in inflammation both in the brain [26] and in the adipose tissue [34] of HFD fed rats supplemented with tart cherries. The same animal model was also explored by Seymour and co-workers [62], who reported that the intake of tart cherries reduces retroperitoneal IL-6 and TNF-α mRNA expression, NF-κB activity, and plasma IL-6 and TNF-α concentrations. These results were consistent with our data, demonstrating that certain inflammatory biomarkers were remarkably decreased in cardiac samples of obese rats following tart cherry seed and juice consumption.

Since diverse study designs have been carried out, in human or animal models, and at many dosages of tart cherry, the results are quite variable. Beneficial effects of tart cherry supplementations were found in female rats, protecting early age-related bone loss and increasing bone mineralization [73]; therefore, it would be interesting to extend the study to female rats. Aside from our current results in male rats, there is wide evidence that demonstrates that tart cherry ingestion provides anti-inflammatory and anti-oxidative activities both in vivo and in vitro [63,67,74,75,76,77,78,79].

## 5. Conclusions

Our findings support the dual antioxidant and anti-inflammatory actions of tart cherries, which may represent an interesting therapeutic strategy to provide a dietary supplement for people either at high risk or with established obesity-related cardiovascular disease.

## Figures and Tables

**Figure 1 biology-11-00646-f001:**
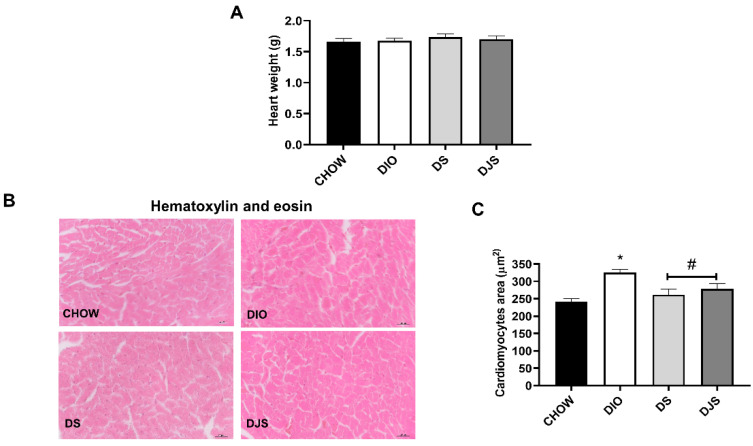
Heart weight and assessment of cardiomyocyte hypertrophy. (**A**) Whole heart weight; (**B**,**C**) Representative pictures of hematoxylin and eosin staining and quantification of cardiomyocyte cross-sectional area CHOW rats (*n* = 8), fed with standard diet; DIO rats (*n* = 9), fed with high-fat diet; DS (*n* = 12), DIO rats supplemented with tart cherry seeds; DJS (*n* = 9), DS rats supplemented with tart cherry juice. Magnification 40×. Scale bar 25 µm. Data are mean ± SEM. * *p* < 0.05 vs. CHOW rats; # *p* < 0.05 vs. DIO rats.

**Figure 2 biology-11-00646-f002:**
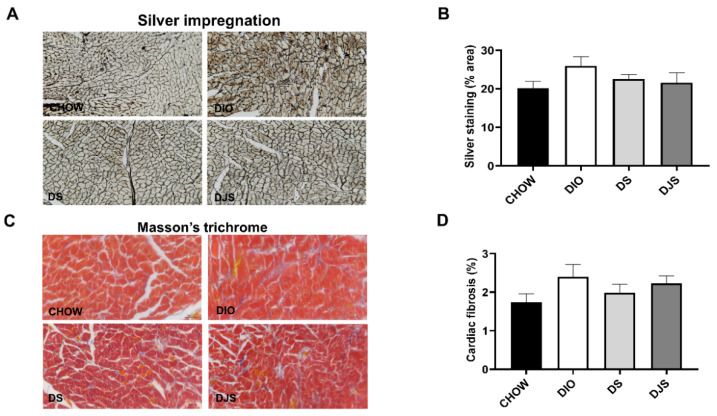
Assessment of cardiac fibrosis. (**A**,**B**) Representative pictures of silver impregnation stained cardiac sections and quantification of silver staining area. Magnification 20×. Scale bar 50 µm; (**C**,**D**) Representative pictures of Masson’s trichrome staining and quantification of fibrosis. Magnification 40×. Scale bar 25 µm. CHOW rats (*n* = 8), fed with standard diet; DIO rats (*n* = 9), fed with high-fat diet; DS (*n* = 12), DIO rats supplemented with tart cherry seeds; DJS (*n* = 9), DS rats supplemented with tart cherry juice. Data are mean ± SEM.

**Figure 3 biology-11-00646-f003:**
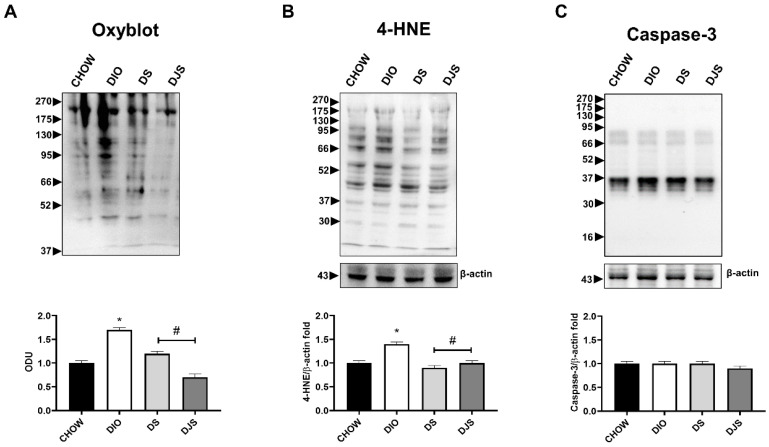
Oxidative stress and apoptosis. (**A**) Oxyblot in cardiac samples and the graph shows the measurements of optical density expressed as arbitrary optical density unit (ODU). Cardiac lysates were immunoblotted with anti-4-Hydroxynonenal (4-HNE) (**B**) and anti-Caspase-3 (**C**). Graphs show the densitometric ratios of bands and β-actin expression, used to normalize the data. CHOW rats (*n* = 8), reference group fed with standard diet; DIO rats (*n* = 9), fed with high-fat diet; DS (*n* = 12), DIO rats supplemented with tart cherry seeds; DJS (*n* = 9), DS rats supplemented with tart cherry juice. Data are mean ± SEM. * *p* < 0.05 vs. CHOW rats; # *p* < 0.05 vs. DIO rats.

**Figure 4 biology-11-00646-f004:**
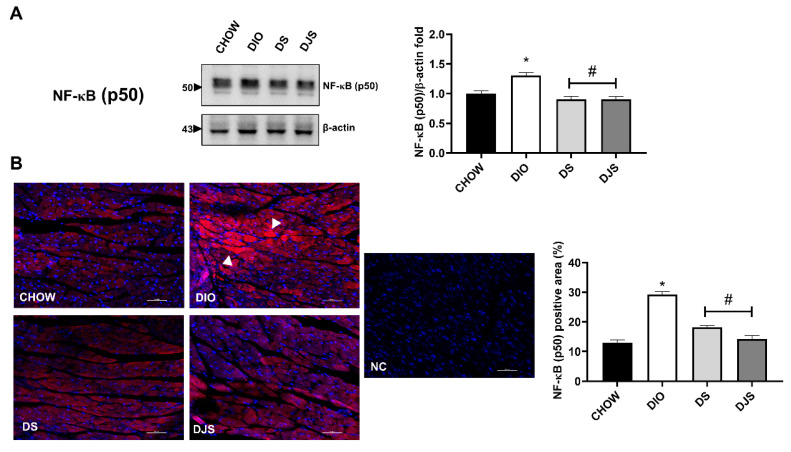
Measurement of nuclear factor kappa-light-chain-enhancer of activated B cells subunit p50 (NF-κB p50). (**A**) Cardiac lysates from rats were immunoblotted using specific anti NF-κB (p50). Graph shows the ratio of densitometric analysis of bands and β-actin expression used to normalize the data, taking CHOW rats as a reference group; (**B**) Confocal image of representative immunofluorescent staining for NF-κB (p50) in the heart and quantification expressed as percentage (%) of NF-κB (p50) positive area. Magnification 10× zoom 3. Scale bar 10 µm. NC, Negative control. Arrowheads indicate the more immunoreactive cardiomyocytes. CHOW rats (*n* = 8), fed with standard diet; DIO rats (*n* = 9), fed with high-fat diet; DS (*n* = 12), DIO rats supplemented with tart cherry seeds; DJS (*n* = 9), DS rats supplemented with tart cherry juice. Data are mean ± SEM. * *p* < 0.05 vs. CHOW rats; # *p* < 0.05 vs. DIO rats.

**Figure 5 biology-11-00646-f005:**
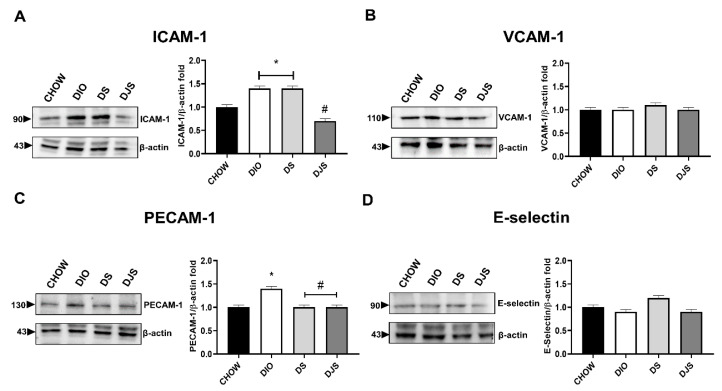
Western blot analysis of inflammatory adhesion molecules. Cardiac lysates were immunoblotted using antibodies against intracellular adhesion molecule-1 (ICAM-1) (**A**); vascular cell adhesion molecule-1 (VCAM-1) (**B**); platelet endothelial cell adhesion molecule-1 (PECAM-1) (**C**); E-selectin (**D**). Graphs show the densitometric ratios of bands and β-actin expression used to normalize the data. CHOW rats (*n* = 8), reference group fed with standard diet; DIO rats (*n* = 9), fed with high-fat diet; DS (*n* = 12), DIO rats supplemented with tart cherry seeds; DJS (*n* = 9), DS rats supplemented with tart cherry juice. Data are mean ± SEM. * *p* < 0.05 vs. CHOW rats; # *p* < 0.05 vs. DIO rats.

**Figure 6 biology-11-00646-f006:**
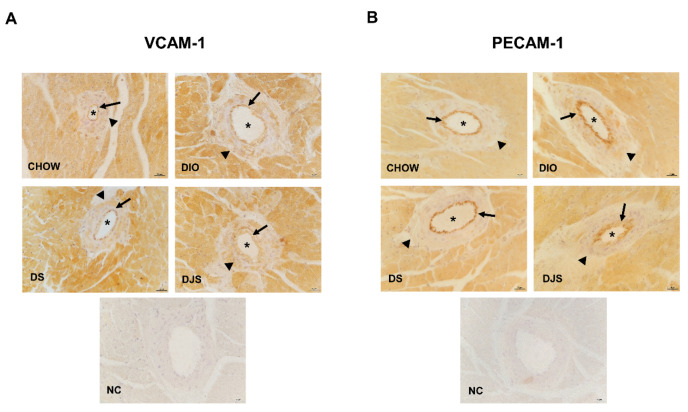
Immunohistochemical analysis of inflammatory adhesion molecules. Representative pictures of heart sections processed for the immunohistochemistry of vascular cell adhesion molecule-1 (VCAM-1) (**A**) and platelet endothelial cell adhesion molecule-1 (PECAM-1) (**B**). The immunoreaction is located in the endothelium (arrows) while the tunica media (arrowheads) is negative. The lumen of vessels is marked with an asterisk (*). Magnification 40×. Scale bar 25 µm. NC, Negative control. CHOW rats (*n* = 8), fed with standard diet; DIO rats (*n* = 9), fed with high-fat diet; DS (*n* = 12), DIO rats supplemented with tart cherry seeds; DJS (*n* = 9), DS rats supplemented with tart cherry juice.

**Figure 7 biology-11-00646-f007:**
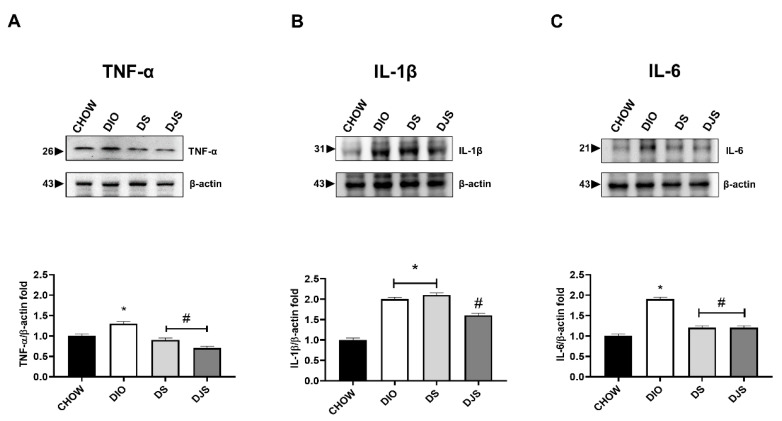
Western blot analysis of inflammatory cytokines. Cardiac lysates were immunoblotted using antibodies against tumor necrosis factor-α (TNF-α) (**A**), anti-interleukin-1β (IL-1β) (**B**), and interleukin-6 (IL-6) (**C**). Graphs show the densitometric ratios of bands and β-actin expression used to normalize the data. CHOW rats (*n* = 8), reference group fed with standard diet; DIO rats (*n* = 9), fed with high-fat diet; DS (*n* = 12), DIO rats supplemented with tart cherry seeds; DJS (*n* = 9), DS rats supplemented with tart cherry juice. Data are mean ± SEM. * *p* < 0.05 vs. CHOW rats; # *p* < 0.05 vs. DIO rats.

**Figure 8 biology-11-00646-f008:**
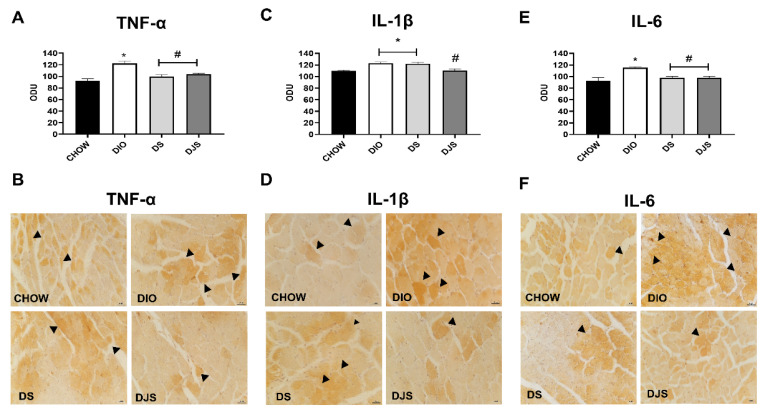
Immunohistochemical analysis of inflammatory cytokines. Graphs indicate the mean intensities of area immunoreaction of tumor necrosis factor-α (TNF-α) (**A**), anti-interleukin-1β (IL-1β) (**C**), and interleukin-6 (IL-6) (**E**) measured in the arbitrary optical density unit (ODU). Data are mean ± SEM. * *p* < 0.05 vs. CHOW rats; # *p* < 0.05 vs. DIO rats. Immunohistochemical representative pictures of heart sections processed for TNF-α (**B**), IL-1β (**D**), and IL-6 (**F**). The immunoreactive cardiomyocytes are indicated with arrowheads. Magnification 40×. Scale bar 25 µm. CHOW rats (*n* = 8), fed with standard diet; DIO rats (*n* = 9), fed with high-fat diet; DS (*n* = 12), DIO rats supplemented with tart cherry seeds; DJS (*n* = 9), DS rats supplemented with tart cherry juice.

## Data Availability

Not applicable.

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
