# Peer review of "Anti-Inflammatory and Antioxidant Properties of Tart Cherry Consumption in the Heart of Obese Rats"

_biology, 2022, doi:10.3390/biology11050646_

Round 1

Reviewer 1 Report

The manuscript by Martinelli et al. highlights the beneficial roles of tart cherry seeds powder and seeds powder plus tart cherry juice on the limitation of high fat diet-induced cardiac inflammation and oxidative stress. The data presented are convincing and bring new evidence of the health benefits of natural compounds. However, some additional experiments would further improve the quality of the paper.

Major comments:

1) Figure 1: Masson’s trichrome staining is a good method to assess fibrosis but it does not provide accurate results to measure cardiomyocytes area. Please perform another technique to quantify cardiomyocytes hypertrophy properly (for instance Vinculin staining).

2) Figure 1: Even if the amounts of fibrosis are low, please provide quantifications of silver impregnation and Masson’s trichrome staining.

3) Figure 4: Immunofluorescence staining of inflammatory cells on cardiac tissue would strengthen all data on inflammation. Please perform CD45 staining on heart sections.

4) Figure 7: Why did the authors provide illustrative pictures for TNF-a and IL-6, but not IL1-B? Illustrative pictures for IL1-B staining would be appreciated too.

Minor comments:

1) M&M: Please provide the detailed composition of the lysis buffer used for western blots

2) M&M: Please provide the detailed references for the antibodies used. Does Caspase 3 antibody bind to both full length and cleaved caspase 3?

3) Do the authors have any evidence that DS and/or DJS benefits on inflammation and oxidative stress also improve cardiac systolic/diastolic function? A comment should be done in the discussion.

4) From a mechanistic point of view, do the authors have evidence that DS and/or DJS impact mitochondrial function?

5) abstract: Please replace “proteins’ expressions” by “protein expressions”

6) Line #48: Please, replace “chronic low-grade of inflammation is an central (...)” by “chronic low-grade inflammation is a central (...)”

Author Response

Reply Reviewer1

Major comments:

1) Figure 1: Masson’s trichrome staining is a good method to assess fibrosis but it does not provide accurate results to measure cardiomyocytes area. Please perform another technique to quantify cardiomyocytes hypertrophy properly (for instance Vinculin staining).

Thank you very much for the suggestion. Unfortunately, we don't have the Vinculin staining, but to meet the reviewer’s request, we performed on paraffin sections the hematoxylin & eosin staining, as by the listed refs. You can find below that H&E staining is a suitable method to measure cardiomyocyte cross-sectional areas.

-Miller EJ, Calamaras T, Elezaby A, Sverdlov A, Qin F, Luptak I, Wang K, Sun X, Vijay A, Croteau D, Bachschmid M, Cohen RA, Walsh K, Colucci WS. Partial Liver Kinase B1 (LKB1) Deficiency Promotes Diastolic Dysfunction, De Novo Systolic Dysfunction, Apoptosis, and Mitochondrial Dysfunction With Dietary Metabolic Challenge. J Am Heart Assoc. 2015 Dec 31;5(1):e002277. doi: 10.1161/JAHA.115.002277.

-Liu W, Zi M, Jin J, Prehar S, Oceandy D, Kimura TE, Lei M, Neyses L, Weston AH, Cartwright EJ, Wang X. Cardiac-specific deletion of mkk4 reveals its role in pathological hypertrophic remodeling but not in physiological cardiac growth. Circ Res. 2009 Apr 10;104(7):905-14. doi: 10.1161/CIRCRESAHA.108.188292.

-Liu W, Zi M, Naumann R, Ulm S, Jin J, Taglieri DM, Prehar S, Gui J, Tsui H, Xiao RP, Neyses L, Solaro RJ, Ke Y, Cartwright EJ, Lei M, Wang X. Pak1 as a novel therapeutic target for antihypertrophic treatment in the heart. Circulation. 2011 Dec 13;124(24):2702-15. doi: 10.1161/CIRCULATIONAHA.111.048785.

-Qin F, Siwik DA, Pimentel DR, Morgan RJ, Biolo A, Tu VH, Kang YJ, Cohen RA, Colucci WS. Cytosolic H2O2 mediates hypertrophy, apoptosis, and decreased SERCA activity in mice with chronic hemodynamic overload. Am J Physiol Heart Circ Physiol. 2014 May 15;306(10):H1453-63. doi: 10.1152/ajpheart.00084.2014.

The reference 37, “Qin et al., 2014”, has been cited in the materials and methods paragraph of the manuscript to provide our analyses. In the current version, we added in the new fig.1 the H&E representative pics, and we measured again the cardiomyocyte cross-sectional areas. As shown by graph (C), there are no great differences between the previous analysis carried out in Masson’s trichrome stained sections. We agree with the reviewer concerning the fact that Masson's trichrome method is used to examine interstitial fibrosis, and we quantified the fibrosis areas (expressed as a %) at 40x magnification. In fig.2, we replace the previous images with the new ones, captured with a higher resolution at 40x.

2) Figure 1: Even if the amounts of fibrosis are low, please provide quantifications of silver impregnation and Masson’s trichrome staining.

Thank you very much for the suggestion. As said before, we quantified in the Masson’s trichrome stained pics the % of the area occupied by fibrosis, so we added the quantification graphs to the new fig.2 (D). The quantification was also done for the silver impregnation staining area (%), which is shown in fig.2 (B).

3) Figure 4: Immunofluorescence staining of inflammatory cells on cardiac tissue would strengthen all data on inflammation. Please perform CD45 staining on heart sections.

We thank the reviewer for the suggestion. Based on literature, we found that the total number of T lymphocytes (CD45+CD3+) was unaffected in db/db mouse cardiac tissue (Ref. Niderla-Bielińska J, Ścieżyńska A, Moskalik A, Jankowska-Steifer E, Bartkowiak K, Bartkowiak M, Kiernozek E, Podgórska A, Ciszek B, Majchrzak B, Ratajska A. A Comprehensive miRNome Analysis of Macrophages Isolated from db/db Mice and Selected miRNAs Involved in Metabolic Syndrome-Associated Cardiac Remodeling. Int J Mol Sci. 2021 Feb 23;22(4):2197. doi: 10.3390/ijms22042197). In addition, in C57BL/6J mice with high fat diet (HFD), there were no significant differences in the proportion of CD45+ CD11b− cells from spleen and peripheral lymph nodes (PLN) in different groups of mice. These data indicated that HFD induced more severe inflammation in central nervous system, but not in systemic level (Ji Z, Wu S, Xu Y, Qi J, Su X, Shen L. Obesity Promotes EAE Through IL-6 and CCL-2-Mediated T Cells Infiltration. Front Immunol. 2019;10:1881. Published 2019 Aug 27. doi:10.3389/fimmu.2019.01881). Moreover, a 3.5-week period of HFD caused a marked increase in CD45+ leukocytes in the heart, but the data was not significant as shown in the Luk’s paper (Luk FS, Kim RY, Li K, Ching D, Wong DK, Joshi SK, Imhof I, Honbo N, Hoover H, Zhu BQ, Lovett DH, Karliner JS, Raffai RL. Immunosuppression With FTY720 Reverses Cardiac Dysfunction in Hypomorphic ApoE Mice Deficient in SR-BI Expression That Survive Myocardial Infarction Caused by Coronary Atherosclerosis. J Cardiovasc Pharmacol. 2016 Jan;67(1):47-56. doi: 10.1097/FJC.0000000000000312).

Nevertheless, we performed the immunofluorescence for CD45 on the heart sections, as the reviewer requested. The CD45+ positive cells were labelled in red (Alexa Fluor 594). Here, we show the representative pictures, and there are no differences between lean and obese rats.

4) Figure 7: Why did the authors provide illustrative pictures for TNF-a and IL-6, but not IL1-B? Illustrative pictures for IL1-B staining would be appreciated too.

Thank you very much for the suggestion. We added the representative pictures for IL1-B in fig.8 (D).

Minor comments:

1) M&M: Please provide the detailed composition of the lysis buffer used for western blots

Thank you. We added in M&M: “Samples were lysed in lysis buffer (1M Tris pH 7.4, 1 M NaCl, 10 mM EGTA, 100 mM NaF, 100 mM Na3VO4, 100 mM PMSF, 2% deoxycholate, 100 mM EDTA, 10% Triton X-l00, 10% glycerol, 10% SDS, 0.1 M Na4P207), containing protease inhibitor cocktail (Sigma-Aldrich), by using a Mixer Mill MM300 (Qiagen, Hilden, Germany)”.

2) M&M: Please provide the detailed references for the antibodies used. Does Caspase 3 antibody bind to both full length and cleaved caspase 3?

Thank you very much for the suggestion. We added the catalog number of antibodies. Concerning the caspase 3: Caspase-3 Antibody #9662 from Cell Signaling. For further details, please see the website: https://www.cellsignal.com/products/primary-antibodies/caspase-3-antibody/9662. Caspase-3 Antibody detects endogenous levels of full length caspase-3 (35 kDa) and the large fragment of caspase-3 resulting from cleavage (17 kDa).

3) Do the authors have any evidence that DS and/or DJS benefits on inflammation and oxidative stress also improve cardiac systolic/diastolic function? A comment should be done in the discussion.

We thank the reviewer for the observation. We added in the discussion section this sentence: “For instance, we have already reported that the DS and DJS rats showed a significant reduction of systolic blood pressure compared with DIO rats [26,33-34]. This effect has been attributed to anthocyanins, which exhibit several biological effects, including vasodilatory capacity. Indeed, it has been reported that the nitric oxide (NO) system may be responsible for the relaxation response of coronary arteries to red fruit extracts [64], and anthocyanins condensed tannin-containing fractions showed more vasodilation property than other polyphenols [17]. In hypertensive stroke-prone rats treated with anthocyanins rich blueberries, it was demonstrated a significant reduction in systolic blood pressure. Anthocyanins exhibited NO-dependent vasodilation via endothelium induced by acetylcholine through the NO metabolism pathway. In addition, the other vasodilator outcome of anthocyanins is the endothelium-dependent relaxation [64,65]. In clinical trials, the sour cherry reduced hypertension and these effects were due to the circulating phenolic content of the cherries [66]. Moreover, Chai et al. [67] suggested that the anti-hypertensive ability of sour cherry can be due to its anti-oxidant and anti-inflammatory activity” Lines 354-368.

4) From a mechanistic point of view, do the authors have evidence that DS and/or DJS impact mitochondrial function?

Thank you very much for the question, but we do not study in the current manuscript if the supplementation could influence the mitochondrial functions. In the discussion, we reported a citation as follows: “4-HNE damages the myocardium interfering with mitochondria and making adducts.”

5) abstract: Please replace “proteins’ expressions” by “protein expressions”

Thank you for the correction. Done.

6) Line #48: Please, replace “chronic low-grade of inflammation is an central (...)” by “chronic low-grade inflammation is a central (...)”

Thank you for the correction. Done.

Reviewer 2 Report

In the current manuscript, Martinelli et al. investigate the cardioprotective effect of tart cherry seeds powder and juice in rats with high-fat diet (HFD) challenge. The authors demonstrate that the tart cherry supplement diminished HFD-induced oxidative stress and inflammation in rat hearts. For the oxidative stress, less HFD-induced cardiac protein 4-hydroxynonenal and carbonyl group modifications were observed in rats with tart cherry supplement. Also, HFD-induced cardiac protein expressions of NF-κB p50 (immunoblotting and immunostaining), adhesion molecules ICAM-1 and PECAM-1 (immunoblotting), and inflammatory chemokines TNF-α, IL-1β, and IL-6 (immunoblotting and immunostaining) were decreased in rats supported with tart cherry products. These results are interesting, but the writing needs to be improved. In addition, some specific issues are listed below.       

1. The study is mainly focus on heart but it is not included in the title.

2. In Abstract, no need to include some abbreviations (e.g. HFD, CHOW) that appear only once. No definition for DIO. The authors describe "The systolic blood pressure, glycemia, and triglycerides were diminished in DS and DJS rats compared to DIO animals." but no data in the manuscript.

3. The dose/concentration of tart cherry seeds powder and juice should be included in Materials and Methods. If the efficient dose in rats can be directly applied to humans? 

4. No sample numbers (n per group) for quantification results.

5. Only male rats were studied in the current manuscript, is the tart cherry supplement beneficial to females? 

6. Some uncropped immunoblots have many bands other than target proteins, it is better to optimize the conditions so the data will be more confident.

7. Majority of error bars in figures look similar, please double check this. 

8. For the NFκB activation, more p50 expression is hard to address its activity, how about to test phospho-p65 S536?

9. What are the potential effectives in the seeds powder and juice from tart cherries to protect heart? What is/are the possible underlying mechanism(s)? 

Author Response

Reply Reviewer 2

  1. The study is mainly focus on heart but it is not included in the title.

We agree, and we specified, in the title, that our study has been carried out in the heart: in the heart of obese rats.

  1. In Abstract, no need to include some abbreviations (e.g. HFD, CHOW) that appear only once. No definition for DIO. The authors describe "The systolic blood pressure, glycemia, and triglycerides were diminished in DS and DJS rats compared to DIO animals." but no data in the manuscript.

Thank you very much for the suggestion. We did not include the abbreviations HFD, CHOW, DS, DJS and 4HNE (once cited), but we specified the definition of DIO as follow in the abstract: “in the heart of diet-induced obese (DIO) rats”. The blood parameters have been already published before, so we prefer to delete that sentence in the abstract. However, these data have been reported (refs. 26,32-35) both in the results as well as in the discussion to reinforce the benefits of tart cherries intake.

  1. The dose/concentration of tart cherry seeds powder and juice should be included in Materials and Methods. If the efficient dose in rats can be directly applied to humans? 

Thank you for the suggestion. We cited our previous studies in order to avoid redundancy. Nevertheless, we added in the materials and methods the dose/concentration of tart cherry seeds powder and juice as follow: ”rats with supplementation of Prunus Cerasus L. 0.1 mg/g/day of tart cherry seed powder (DS rats; n = 12), and rats with supplementation of Prunus Cerasus L. 0.1 mg/g/day of tart cherries seeds powder plus tart cherry juice, containing 1 mg of anthocyanins (DJS rats; n = 9)”. These concentrations were chosen because they are in the range with activity or effectivity and no toxicity based on previous animals studies present in the literature (Lee et al. J AOAC Int. 2005, 88, 1269-1278; Bak et al. Am. J. Physiol. Heart. Circ . Physiol.2006, 291, H1329-1336; Bak et al. Phytother Res.2011, 25, 1714-1720). We quantified that the same amount of tart cherries seeds gave to the animals is around 1 g for human. Instead, regarding the juice in human, we estimated approximately 60 ml of juice containing 30 mg of anthocyanins.    

  1. No sample numbers (n per group) for quantification results.

Thank you very much for the observation. For the revision, we have now specified in each figure legends the number of rats for each group. We would like to add that: “We excluded n=6 of the 36 rats fed with HFD from the study, because they did not significantly increase body weight compared to CHOW rats [26]”. Final amounts of animals analyzed: CHOW (n=8), rats fed with standard diet; DIO (n=9), rats fed with HFD; DS (n=12), DIO rats supplemented with tart cherry seeds; DJS (n=9), DS rats supplemented with tart cherry juice.

  1. Only male rats were studied in the current manuscript, is the tart cherry supplement beneficial to females? 

Thank you for the suggestion. We used the male rats only to avoid hormonal interference on all parameters measured. Moreover, in male rats the diabetes develops at 8 weeks of age, while females do not develop evident diabetes (Peterson et al., 1990, ILAR J. 32, 16–19; Srinivasan and Ramarao,2007;Indian J. Med. Res. 125, 451–472; Wohlfart et al., 2014 ActaDiabetol. 51, 553–558). However, we added in the discussion that it would be interesting to extend the study also to female rats and we cited a study reporting the intake of tart cherries seems to be beneficial in female animals by Smith et al., 2019 (Added Reference Smith BJ, Crockett EK, Chongwatpol P, Graef JL, Clarke SL, Rendina-Ruedy E, Lucas EA. Montmorency tart cherry protects against age-related bone loss in female C57BL/6 mice and demonstrates some anabolic effects. Eur J Nutr. 2019 Dec;58(8):3035-3046. doi: 10.1007/s00394-018-1848-1).

  1. Some uncropped immunoblots have many bands other than target proteins, it is better to optimize the conditions so the data will be more confident.

Thank you very much for the observation. WB protocols for each antibody have been already set up from preliminary experiments (Please, see the ref. 36. Martinelli, I.; Tomassoni, D.; Moruzzi, M.; Roy, P.; Cifani, C.; Amenta, F.; Tayebati, S.K. Cardiovascular Changes Related to Metabolic Syndrome: Evidence in Obese Zucker Rats. Int J Mol Sci. 2020, 21, 2035. doi: 10.3390/ijms21062035). In the cited manuscript, the same antibodies were immunoblotted in the heart of male rats. To replicate the working conditions, obviously also the antibodies dilutions were the same. As you found in the file of the entire gels, no unspecific binding sites were present for the antibodies purchased by Cell Signaling or Sigma, differentially from those by Santacruz biotechnology. Among the WB images, PECAM is the one with many bands other than target proteins. Concerning the WB procedures, we used to block with BSA 5%, followed by best range of working dilution, and to select the band of molecular weight specified in the datasheet of the company. To satisfy reviewer’s request, we repeated the immunoblotting for PECAM-1, trying to optimize the protocols. We have tested the blocking solution with milk 5%, as well as antibody diluted in milk 3%. As showed by the representative figure. Below, the unspecific binding sites were a lit bit reduced, but unfortunately, these protocol changes covered the specific band at 130 kDa (really faint), corresponding to the correct molecular weight for PECAM. If the reviewer agrees, we would like to keep the previous gel for PECAM.

  1. Majority of error bars in figures look similar, please double check this. 

Thank you very much for the observation. We checked again the raw data and we confirm the error bars.

  1. For the NFκB activation, more p50 expression is hard to address its activity, how about to test phospho-p65 S536?

We thank the reviewer for the suggestion. We justified why we chose NFκB (p50) in the results as follow: “As the NF-κB is a transcription factor responsible for triggering the immune response, and the most prevalent activated form of NF-κB is a heterodimer consisting of a p50 or p52, its expression was investigated.” Moreover, Frantz et al., reported that p50 is crucial for heart failure, after myocardial infarction, and p50 might thus be an ideal target to suppress innate immune activation and to test the hypothesis of a detrimental role for NF-κB activation in congestive heart failure (Frantz S, Hu K, Bayer B, Gerondakis S, Strotmann J, Adamek A, Ertl G, Bauersachs J. Absence of NF-kappaB subunit p50 improves heart failure after myocardial infarction. FASEB J. 2006 Sep;20(11):1918-20. doi: 10.1096/fj.05-5133fje). In addition, p50 NF-kB controls cells survival, tolerogenic as well as immunogenic functions and p50 has been shown to be a major determinant of both innate and adaptive immune responses, underping its relevance in diseases characterized by aberrant immune responses (Larghi P, Porta C, Riboldi E, Totaro MG, Carraro L, Orabona C, Sica A. The p50 subunit of NF-κB orchestrates dendritic cell lifespan and activation of adaptive immunity. PLoS One. 2012;7(9):e45279. doi: 10.1371/journal.pone.0045279). In accordance with our study, the expression of the p50 subunit in heart was found significantly increased in rats fed high fat diet (Jovanovic A, Sudar-Milovanovic E, Obradovic M, Pitt SJ, Stewart AJ, Zafirovic S, Stanimirovic J, Radak D, Isenovic ER. Influence of a High-Fat Diet on Cardiac iNOS in Female Rats. Curr Vasc Pharmacol. 2017;15(5):491-500. doi: 10.2174/1570161114666161025101303), and in a rat model of myocardial infarction, the NF-kB p50 was significantly increased, but the anti-inflammatory properties of atorvastatin treatment significantly reduced the mRNA expression level of the NF-κB p50 compared to the control group, but without affecting the expression of the NF-κB p65 (Reichert K, Pereira do Carmo HR, Galluce Torina A, Diógenes de Carvalho D, Carvalho Sposito A, de Souza Vilarinho KA, da Mota Silveira-Filho L, Martins de Oliveira PP, Petrucci O. Atorvastatin Improves Ventricular Remodeling after Myocardial Infarction by Interfering with Collagen Metabolism. PLoS One. 2016 Nov 23;11(11):e0166845. doi: 10.1371/journal.pone.0166845).

  1. What are the potential effectives in the seeds powder and juice from tart cherries to protect heart? What is/are the possible underlying mechanism(s)? 

We thank the reviewer for the observation. We added in the discussion section this sentence: “For instance, we have already reported that the DS and DJS rats showed a significant reduction of systolic blood pressure compared with DIO rats [26,33-34]. This effect has been attributed to anthocyanins, which exhibit several biological effects, including vasodilatory capacity. Indeed, it has been reported that the nitric oxide (NO) system may be responsible for the relaxation response of coronary arteries to red fruit extracts [64], and anthocyanins condensed tannin-containing fractions showed more vasodilation property than other polyphenols [17]. In hypertensive stroke-prone rats treated with anthocyanins rich blueberries, it was demonstrated a significant reduction in systolic blood pressure. Anthocyanins exhibited NO-dependent vasodilation via endothelium induced by acetylcholine through the NO metabolism pathway. In addition, the other vasodilator outcome of anthocyanins is the endothelium-dependent relaxation [64,65]. In clinical trials, the sour cherry reduced hypertension and these effects were due to the circulating phenolic content of the cherries [66]. Moreover, Chai et al. [67] suggested that the anti-hypertensive ability of sour cherry can be due to its anti-oxidant and anti-inflammatory activity. Interestingly, the oleic and linoleic acids, as main components of tart cherry seeds, have been shown to protect the endothelium [68]. These compounds could explain the anti-inflammatory effects also observed in the brain of DIO rats. A diet rich in oleic acid can have a beneficial effect on diabetes, and can reverse the negative effects of inflammatory cytokines found in obesity and diabetes [69]. Oleic acid has been suggested to protect against cardiovascular insulin resistance, improving endothelial dysfunction in response to proinflammatory signals, and reducing proliferation and apoptosis in vascular smooth muscle cells. These effects may contribute to plaque stability and an ameliorated atherosclerotic process [70]. Furthermore, it has been reported by the intake of sour cherry seed kernel extract im-proved postischemic recovery of cardiac function during reperfusion. Beside other potential action mechanisms of proanthocyanidin, trans-resveratrol, and flavonoid components of the extract, could be responsible for the cardioprotection in ischemic-reperfused myocardium [71]. Finally, preclinical evidence showed benefits in dietary supplementation of the cited above antioxidant compounds on fat accumulation [11]. Lines 354-382.

Round 2

Reviewer 1 Report

The authors have taken into consideration the remarks. Additional experiments and corrections have been done and the paper has been greatly improved.

Author Response

We thank the reviewer for his suggestions.

Reviewer 2 Report

The current manuscript is largely improved. A minor suggestion is to provide more detail title in figure legend (Figure 1. Weight and heart morphology vs Figure 2. Heart morphology; Figure 4, 5, 6, 7, and 8 all Inflammation). 

Author Response

We thank the reviewer for the corrections. We have changed the legend titles in the manuscript as follow:

Figure 1. Heart weight and assessment of cardiomyocytes hypertrophy.

Figure 2. Assessment of cardiac fibrosis.

Figure 4. Measurements of nuclear factor kappa-light-chain-enhancer of activated B cells subunit p50 (NF-κB p50).

Figure 5. Western blot analysis of inflammatory adhesion molecules.

Figure 6. Immunohistochemical analysis of inflammatory adhesion molecules

Figure 7. Western blot analysis of inflammatory cytokines.

Figure 8. Immunohistochemical analysis of inflammatory cytokines.